# Tpeak-Tend Interval during Pregnancy and Postpartum

**DOI:** 10.3390/ijerph191912638

**Published:** 2022-10-03

**Authors:** Tomasz Kandzia, Grażyna Markiewicz-Łoskot, Przemysław Binkiewicz

**Affiliations:** 1Department of Nursing and Social Medical Problems, Faculty of Health Sciences in Katowice, Medical University of Silesia, 40-752 Katowice, Poland; 2Department of Gynaecology and Obstetrics in Pyskowice, 44-120 Pyskowice, Poland

**Keywords:** pregnancy, electrocardiography, electrophysiology

## Abstract

Background: Pregnancy is a condition in which new cardiac arrhythmias can occur or prior undiagnosed arrhythmias may provide symptoms. The occurrence of severe ventricular arrhythmias and polymorphic ventricular tachycardia that may lead to fainting or sudden cardiac death is promoted by the prolongation of the QTc interval. The post-partum adaptation period is the most arrhythmogenic. TpTe (Tpeak-Tend interval) is a novel marker of arrhythmogenesis by many considered a more sensitive marker than QTc. Objective: The aim of our work was to determine the TpTe interval (Tpeak-Tend) in women in the first, second and third trimester of pregnancy and the post-partum period. Materials and Methods: The study group consisted of 128 women in pregnancy or postpartum and a control group of 32 non-pregnant women. A standard 12-lead ECG (electrocardiograph) recording with evaluation of the duration of TpTe and QTc was performed in all patients. Results: In comparison to the non-pregnant women, higher values of QTc and TpTe were observed starting in the first trimester with highest values observed in the postpartum period. Mean duration of TpTe interval during pregnancy (81.59 ± 5.92 ms) and in the whole study group (pregnancy + postpartum) (85.46 ± 6.45 ms) was significantly longer (*p* < 0.001) compared to the TpTe interval in the control group (74.06 ± 6.14 ms). During pregnancy and postpartum, the increase in the TpTe interval in comparison to the increase in the QTc parameter (31.10% vs. 4.18%) was significantly higher (*p* < 0.001). Conclusions: The study showed an increase in the duration of the TpTe interval and QTc parameter during pregnancy and postpartum with the highest values in the postpartum period. TpTe interval increase was significantly higher compared to QTc increase during pregnancy and postpartum. Changes of TpTe interval were not associated with any clinical outcome or measure of arrythmia burden. Further studies are needed in order to see the clinical significance of these ECG findings, in particular for larger groups of patients with automatic measurement in correlation with echocardiography

## 1. Introduction

Numerous hemodynamic changes, in particular the stretching of the atrial and ventricular muscle fibers, increase the risk of arrhythmogenesis during pregnancy and the postpartum. In the United States over the past two decades, an increasing maternal mortality and incidence of arrhythmias in pregnancy have been noted. Pregnancy is associated with a greater risk of arrhythmias, and patients with a history of arrhythmias are at a significant risk of arrhythmia recurrence during pregnancy [1]. Increase in arrhythmogenesis during pregnancy also affects women with no previous positive cardiological history. For every third pregnant woman with an arrhythmia detected during pregnancy, it is her first attack of arrhythmia [2]. The occurrence of dangerous ventricular arrhythmias and polymorphic ventricular tachycardia, which may lead to syncope or sudden cardiac death, is mostly caused by prolongation of the QTc interval [3].

Arrhythmias, especially of ventricular origin, increase the possibility of syncope with the probability of sudden cardiac death [4,5] The postpartum period of adaptive changes in the circulatory system is the most arrhythmogenic [6,7]. The risk of severe ventricular arrhythmias increases in pregnant women with a bad family history (sudden deaths before 30 years of age) or diagnosed with long QT syndrome (LQTS), with a 2.5-fold greater risk in the puerperium for adverse cardiac events compared to pre-pregnancy [8]. In untreated patients, the period of increased risk may persist up to 9 months after delivery [9,10]. There is an increased duration of electrocardiographic parameters of the repolarization period: QT interval and especially the TpTe interval, the most sensitive marker of arrhythmogenesis, along with a change in the shape of the T wave, may predispose to life-threatening ventricular arrhythmias [11]. The assessment of the duration of electrocardiographic parameters of the repolarization period: QT, TpTe (Tpeak-Tend) in a group of women during pregnancy and puerperium, may prove to be diagnostically useful in the early detection of arrhythmias and in the assessment of the increased risk of arrhythmogenesis. Early identification of risk groups will enable the implementation of therapeutic and preventive measures along with a change in lifestyle and pharmacotherapy [12].

The electrocardiographic TpTe interval, reflecting the global dispersion (diffusion) of repolarization, is considered a diagnostic marker of arrhythmogenesis, especially accompanied by a change in the shape of the T wave into two-humid or two-phase [13,14]. Changes in the ventricular repolarization period: QT interval and TpTe allow for assessing the electrical instability of the heart muscle, which may lead to the development of potentially lethal ventricular arrhythmia—polymorphic ventricular tachycardia [15]. The QT interval is determined by duration of the repolarization action potential of the epicardium wall, subendocardium, and M-cells in the deep subendocardium. On the basis of experimental studies performed with the use of transmural cellular electrocardiogram, it has been proved that the longest duration of the repolarization potential occurs in M cells. This potential may be prolonged under the influence of drugs or slow heart rate [16]. The epicardial fibers are repolarized until the end of the T wave. The longest time of repolarization presented by the M cells is recorded in the ECG until the end of the T wave [17]. The earlier termination of repolarization in the epicardium fibers (Q-Tpeak), despite the later arrival of the excitation wave with the simultaneous duration of repolarization in the subendocardium and M cells, creates a transmural, repolarization gradient called repolarization dispersion (scattering), corresponding in the ECG record, TpTe interval [16] (Figure 1).

The increase in transmural repolarization dispersion is probably a substrate for the development of life-threatening ventricular arrhythmias in all ion channel diseases: LQTS, Brugad’s syndrome, short QT syndrome and polymorphic ventricular tachycardia [17,18]. The TpTe interval, reflecting the global dispersion of repolarization, is considered a diagnostic marker of arrhythmogenesis. Variability of electrocardiographic repolarization parameters may prove to be an important prognostic factor in the assessment of threatening, symptomatic ventricular arrhythmia [14,19]. The aim of our study was the assessment of the duration of electrocardiographic parameters of the repolarization period: QTc and TpTe (Tpeak-Tend) in a group of women during pregnancy and puerperium.

## 2. Materials and Methods

The data that support the findings of this study are available from the corresponding author upon reasonable request. The study included a group of 160 women, of whom 128 were during physiological pregnancy or puerperium (study group) and 32 healthy women who came to the clinic for preventive examinations (control group). All patients had a medical history assessment, physical and gynecological examination, laboratory tests, cardiac echocardiography and a standard 12-lead ECG with an assessment of the duration of repolarization parameters: QT and TpTe. In the study and control group, 12-lead standard ECG recordings made before the gynecological and obstetric examination were analyzed. The amplitude and shape of the T wave as well as the duration of individual repolarization parameters were assessed: QT interval and TpTe interval. The study was approved by University of Silesia ethics committee. An informed consent has been acquired from all participants. The study was conducted according to the guidelines of the Declaration of Helsinki. The data presented in this study are available on request from the corresponding author.

Measurements were made manually in the second limb lead (II) and in the fifth precordial lead (V5) on standard 12-lead ECG recordings. The analyzed measurements were the average of three consecutive QRS-T evolutions, with a paper advance of 50 mm/s and a standard feature amplitude—1 mV = 1 cm. From three consecutive QRS-T evolutions, the duration of the repolarization period QT interval and TpTe interval were determined. The total period of QT repolarization was determined from the beginning of the Q wave to the end of the T wave, defined as the point of return of the descending T wave to the isoelectric line excluding the U wave. The QT segment was measured from the beginning of the Q wave to the beginning of the T wave. The period of late TpTe repolarization was calculated from the peak of the T wave to the end of the T wave [20].

Corrected duration of QT versus heart rate was calculated using the Bazett formula (QTcB). In the case of biphasic T waves, the measurements included the first peak of the T wave. All measurements of the end of T wave were analyzed blindly by another independent investigator without access to previously obtained results and clinical data. The highest quality ECGs were analyzed without network disturbances.

## 3. Results

The mean duration of the QTcB increased from the first trimester of pregnancy (420.57 ± 24.91—in the first trimester, 427.58 ± 18.61 ms (milliseconds)—in the second trimester, 426.56 ± 16.12 ms in the third trimester), with the highest mean values observed in the puerperium (428.83 ± 22.52 ms). The lowest mean QTcB values were found in the control group (411.23 ± 20.84 ms) (Figure 2).

QTcB interval was significantly higher in the second and third trimesters and in the puerperium compared to the control group (*p* = 0.003, *p* = 0.003, *p* = 0.002). The QTcB interval was significantly longer (*p* < 0.001) during pregnancy (425.21 ± 19.23 ms) and in the entire study group (pregnancy + puerperium) (426.38 ± 18.92 ms) compared to the control group (411.23 ± 20.84 ms) (Table 1).

Mean TpTe interval increased in the following trimesters of pregnancy (76.11 ± 6.45 ms; 81.33 ± 5.71 ms; 87.33 ± 5.83 ms), reaching the highest value in puerperium (97.09 ± 1.95 ms) (Figure 3). Mean TpTe duration was shortest in the control group (74.06 ± 6.14 ms) with a value range of 60 ms to 80 ms. The maximum values of the TpTe interval were recorded in the puerperium (130 ms).

Mean TpTe interval was significantly longer in the second and third trimesters and in the puerperium compared to the duration of TpTe interval in the control group (*p* < 0.001), with no statistically significant difference between the first trimester and the control group. Mean duration of the TpTe interval during pregnancy (81.59 ± 5.92 ms) and in the entire study group (pregnancy + puerperium) (85.46 ± 6.45 ms) was significantly longer (*p* < 0.001) compared to the TpTe interval in the control group (74.06 ± 6.14 ms) (Table 2).

Comparing the mean durations of QTcB and TpTe intervals in the study and control group, an increase in the duration of QTcB and TpTe interval was found starting in the first trimester with maximal values observed in postpartum (Figure 4). During pregnancy and puerperium, the increase in TpTe interval compared to the increase of QTcB parameter (31.10% vs. 4.18%) was significantly greater (*p* < 0.001).

The ratio of TpTe interval to QT interval (TpTe/QT) showed an increase in the subsequent trimesters of pregnancy (1st trimester—0.21, 2nd trimester—0.22) with maximal values observed in the 3rd trimester and puerperium—0.24. The lowest mean values of the index were found in the control group—0.20, the highest—in the third trimester of pregnancy and in the postpartum period—0.24. The values of the TpTe/QT index showed a statistically significant difference between all groups with no difference in the index values between the third trimester and the postpartum period (*p* = 0.469). The mean values of the TpTe/QT ratio were significantly higher during pregnancy and in the entire study group (pregnancy + puerperium) compared to the TpTe/QT values in the control group (*p* < 0.01) (Figure 5 and Table 3).

## 4. Discussion

The electrocardiographic TpTe interval, reflecting the global dispersion (scattering) of repolarization, is considered to be a more sensitive diagnostic marker of arrhythmogenesis, compared to the traditionally used QT interval, especially with the accompanying change in the shape of the T wave to biphasic [13].

In our own study, a significant extension of the TpTe interval was found in the study group during pregnancy and puerperium (TpTe = 85.12 ms) compared to the control group (74.06 ms). The prolongation of the TpTe interval begins in the first trimester of pregnancy (76.11 ms), reaching the highest values in postpartum period (97.09 ms) with an average value of 23.03 ms. The TpTe interval showed significantly higher values compared to the control group, both during pregnancy (*p* < 0.001) and during pregnancy and puerperium (*p* < 0.001). The increase in the TpTe interval compared to the control group during the first trimester-puerperium period was 31.10%. This is significantly higher compared to the prolongation of the QTcB interval by 4.18% during the same period. The TpTe/QT interval did not show large upward trends in the study group. Our study was limited by the study group count. Larger population studies are needed to determine medical norms for TpTe interval during pregnancy.

Similarly, Tanindi et al. found a gradual increase in the duration of the TpTe interval during pregnancy with a simultaneous significant prolongation of the QT interval. Mean TpTe values in the first trimesters of pregnancy in the control group were 72.7 ms, while in the third trimester, the TpTe interval was 87.2 ms and was similar to the TpTe interval observed in our study (87.3 ms). Tanindi also calculates the TpTe/QT ratio, which was 0.20 in the third trimester (0.24 in our study) [21]. This is the only study in the available literature on changes in the duration of the TpTe interval in women during the physiological period of pregnancy and puerperium. It has been proven that the TpTe interval does not prolong during spinal anesthesia during the cesarean section. On the other hand, administration of oxytocin during cesarean section increases the TpTe interval by an average of 21 ms, which lasts up to 10 min after drug administration, while the QTc interval returns to normal approximately 1 min after drug administration [22].

In pregnant patients with LQTS, higher baseline TpTe interval values (118 ms) were observed compared to our own studies. The TpTe interval was 96 ms in the first trimester of pregnancy with an increase of 16 ms in the 28th week of pregnancy. In the group of women with LQTS who received beta-blocker therapy, the TpTe interval did not extend, shortening on average by 3 ms [8]. In the studied group of women in physiological pregnancy, the average increase in TpTe was 11 ms in the third trimester compared to the first trimester of pregnancy. The diagnostic value of prolongation of the TpTe interval is especially important in the assessment of the risk of preeclampsia, one of the main causes of mortality of patients, fetuses and newborns. In the study conducted by Kirbas et al., the mean TpTe time and the TpTe/QT ratio were significantly higher in the group with moderate and severe preeclampsia compared to the control group of pregnant women in physiological pregnancy, creating an increased risk of ventricular arrhythmias [11]. Moreover, in the group of patients with pregnancy-induced hypertension, longer TpTe intervals were found compared to the control group, consisting of patients in a physiological pregnancy [23]. One of the probable factors responsible for prolongation of the TpTe interval during the pregnancy and puerperium are elevated levels of hormones, mainly estrogen. Cheng et al. observed an extension of the repolarization period in females, both in humans and in animals, which was correlated with increased levels of estrogen in the bloodstream [24]. In addition, in women who underwent virilization, the duration of the TpTe interval was significantly longer compared to the TpTe value in the healthy population of women [25]. Similarly, prolongations in the QT interval and the TpTe interval have been observed in healthy postmenopausal women receiving estrogen-only hormone replacement therapy [26].

The duration of the TpTe interval correlates with the thickness of the left ventricular wall [27,28], which may explain the prolongation of the TpTe interval during pregnancy and the normalization of the TpTe interval depending on the recovery of the left ventricular muscle to pre-pregnancy thickness. The duration of the TpTe interval is gender-dependent, with higher TpTe values in men than in women [24] and increases with age, shortening after 65 years of age. In the study conducted by Braschi et al., the mean duration of the TpTe interval in the adult group (20–65 years of age) was 86.1 ms for women and 90.8 ms for men, with the mean value of the TpTe interval in girls (9–10 years) was 83.6 ms, increasing to adulthood by only 2.5 ms, while in the group of pregnant women, the increase in TpTe within 10 months until the time of puerperium was 23.03 ms. This may indicate very intense electrophysiological changes occurring in women during pregnancy and puerperium [29]. In studies on twin siblings, it was found that the duration of the TpTe interval is constant and inherited [30]. In a study by Wang et al., prolongation of TpTe interval above 120 ms was associated this study group, TpTe interval value of 120 ms or higher occurred in the postpartum period in seven healthy women in physiological pregnancy, with a lower, normal TpTe interval at baseline, and the absence of such prolonged TpTe intervals in the control group. In our study, nine patients showed an increase in the duration of the QTcB interval over 460 ms, whereas six patients had an TpTe interval over 120 ms.

The TpTe interval and TpTe/QT ratio were significantly higher in the Prinzmetal group of patients with polymorphic ventricular arrhythmias compared to the group of patients with non-arrhythmic angina, with the TpTe dispersion index with the odds ratio being the best prognostic factor (OR 1.497) [31]. Similarly, prolongation of the TpTe interval has been shown with a higher incidence of ventricular arrhythmias and the risk of sudden cardiac death in patients with reduced left ventricular ejection fraction [32], in acute coronary syndrome [33], in chronic obstructive pulmonary disease [34] in smokers [35] and in type 2 diabetes [36]. The TpTe interval was a more sensitive marker of threatening ventricular arrhythmia than the standardized QTc interval. In our study, both QTc interval and TpTe interval showed statistically significantly higher values during pregnancy and puerperium than in the control group (*p* < 0.001), but the duration of the TpTe interval showed a 7.44 times higher increase in pregnancy and puerperium compared to the QTc interval.

Similarly, the TpTe/QT ratio is considered a better marker of arrhythmogenesis, especially in left ventricular hypertrophy [27,37,38], than the commonly used corrected QT interval (QTcB). In the study by Ciobanu et al., only the TpTe interval and the TpTe/QT ratio showed a high correlation with left ventricular remodeling and the appearance of premature ventricular beats [39]. During pregnancy, the TpTe/QT ratio is increased in patients with preeclampsia and pregnancy-induced hypertension [11,23]. In our study, the TpTe/QT ratio, like the TpTe interval, was significantly (*p* < 0.001) higher in the third trimester of pregnancy and in puerperium compared to the control group (0.24 vs. 0.20), indicating an increased potential risk of arrhythmogenesis, according to literature [40].

## 5. Conclusions

The duration of the electrocardiographic TpTe interval is significantly extended during pregnancy with highest TpTe values in postpartum, with no significant changes in the QTc interval. Therefore, the assessment of the duration of the TpTe interval may be helpful in identifying pregnant and postpartum women at risk of a potential ventricular arrhythmia (especially in the last trimesters of pregnancy and in the postpartum period), enabling the early implementation of therapeutic and preventive measures, mainly with a history of arrhythmia.

The limitation of our study is that changes of the TpTe interval were not associated with any clinical outcome or measure of arrythmia burden. Further studies are needed in order to see the clinical significance of these ECG findings, in particular for larger groups of patients with automatic measurement in correlation with echocardiography.

## Figures and Tables

**Figure 1 ijerph-19-12638-f001:**
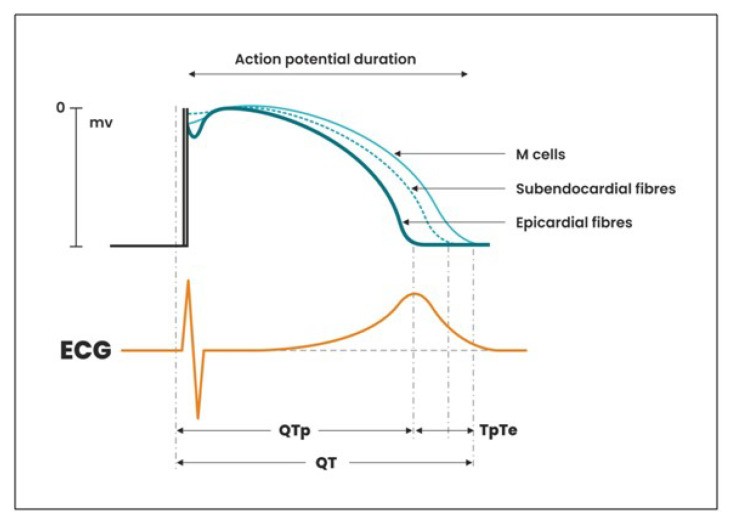
Formation of total QT repolarization time, QTp (Q-Tpeak) and TpTe interval.

**Figure 2 ijerph-19-12638-f002:**
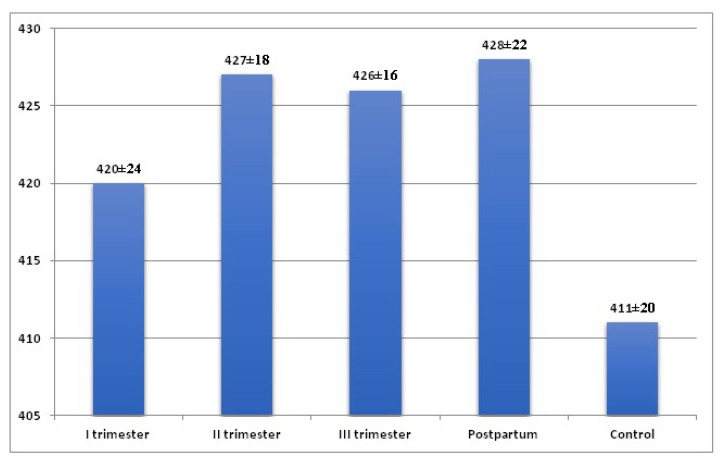
QTc interval in women in the 1st, 2nd, and 3rd trimester of pregnancy, in the puerperium and in the control group.

**Figure 3 ijerph-19-12638-f003:**
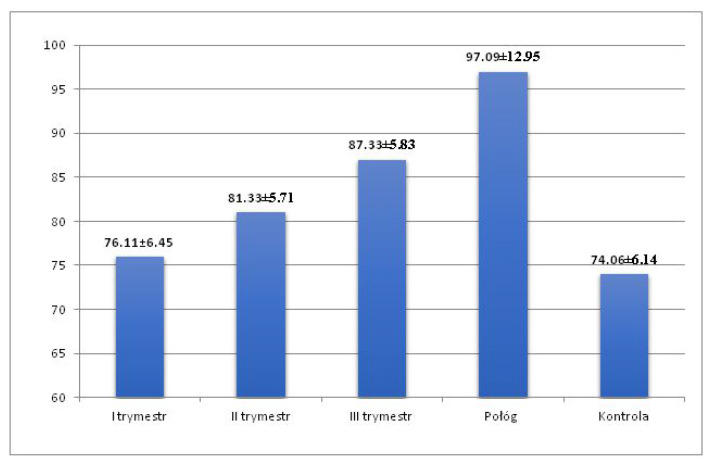
TpTe interval in the 1st, 2nd, and 3rd trimester of pregnancy, in the puerperium and in the control group.

**Figure 4 ijerph-19-12638-f004:**
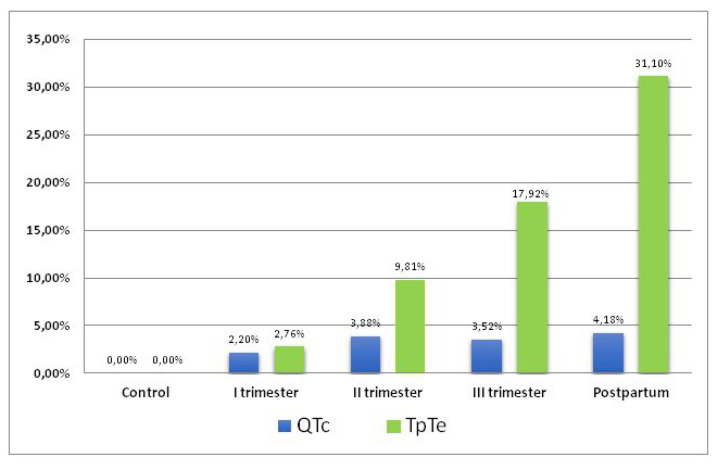
Values of QTcB and TpTe intervals in relation to the control group during pregnancy and puerperium.

**Figure 5 ijerph-19-12638-f005:**
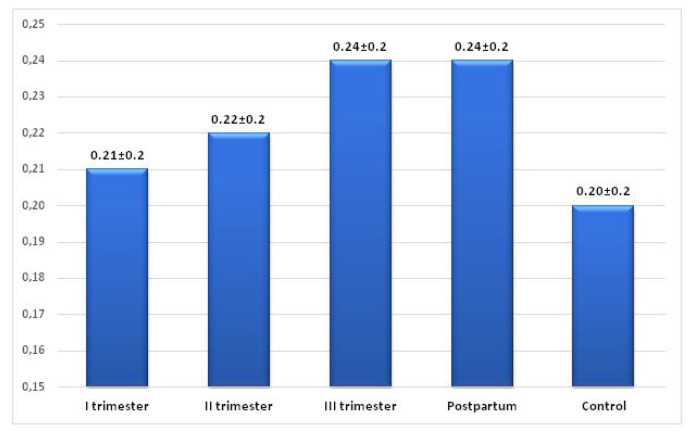
Changes in the TpTe/QT ratio in the study and control group.

**Table 1 ijerph-19-12638-t001:** QTcB in women in the 1st, 2nd and 3rd trimester of pregnancy, in the puerperium and in the control group.

QTcB [ms]	I Trimester	II Trimester	III Trimester	Postpartum	Control
**N**	36	30	30	32	32
**Mean**	420.57	427.58	426.56	428.83	411.23
**SD**	24.91	18.61	16.12	22.52	20.84
**Minimum**	372.62	388.89	390.01	394.13	378.11
**Maximum**	466.16	460.93	466.69	505.96	455.96
**Median**	420.65	427.42	426.82	426.98	411.63
**I trimester**	—	*p* = 0.212	*p* = 0.306	*p* = 0.163	*p* = 0.110
**II trimester**	*p* = 0.212	—	*p* = 0.742	*p* = 0.815	*p* = 0.002
**III trimester**	*p* = 0.306	*p* = 0.742	—	*p* = 0.589	*p* = 0.003
**Postpartum**	*p* = 0.163	*p* = 0.815	*p* = 0.589	—	*p* = 0.002
**Control**	*p* = 0.110	*p* = 0.002	*p* = 0.003	*p* = 0.002	—
**(I-III trim.) vs. Control**				*p* < 0.001	
**(I trim-post.) vs. Control**				*p* < 0.001	

SD—Standard Deviation, Control—Control Group.

**Table 2 ijerph-19-12638-t002:** TpTe interval in the 1st, 2nd, and 3rd trimester of pregnancy, in the puerperium and in the control group.

TpTe	I Trimester	II Trimester	III Trimester	Postpartum	Control
**N**	36	30	30	32	32
**Mean**	76.11	81.33	87.33	97.09	74.06
**SD**	6.45	5.71	5.83	12.95	6.14
**Minimum**	60.00	70,00	80.0	80.00	60.00
**Maximum**	90.00	90.00	100,00	130,00	80,00
**Median**	80.00	80.00	90.00	100.00	70.00
**I trimester**	—	*p* < 0.001	*p* < 0.001	*p* < 0.001	*p* = 0.186
**II trimester**	*p* < 0.001	—	*p* < 0.001	*p* < 0.001	*p* < 0.001
**III trimester**	*p* < 0.001	*p* < 0.001	—	*p* < 0.001	*p* < 0.001
**Postpartum**	*p* < 0.001	*p* < 0.001	*p* < 0.001	—	*p* < 0.001
**Control**	*p* = 0.186	*p* < 0.001	*p* < 0.001	*p* < 0.001	—
**(I-III trim.) vs. Control**				*p* < 0.001	
**(I trim-post.) vs. Control**				*p* < 0.001	

SD—Standard Deviation, Control—Control Group.

**Table 3 ijerph-19-12638-t003:** TpTe/QT ratio in the 1st, 2nd, and 3rd trimester of pregnancy, in the puerperium and in the control group.

TpTe/QT	I Trimester	II Trimester	III Trimester	Postpartum	Control
**N**	36	30	30	32	32
**Mean**	0.21	0.22	0.24	0.24	0.20
**SD**	0.02	0.02	0.02	0.03	0.02
**Minimum**	0.16	0.19	0.20	0.16	0.14
**Maximum**	0.24	0.25	0.27	0.29	0.23
**Median**	0.21	0.22	0.24	0.25	0.20
**I trimester**	—	*p* = 0.012	*p* < 0.001	*p* < 0.001	*p* = 0.045
**II trimester**	*p* = 0.012	—	*p* < 0.001	*p* < 0.001	*p* < 0.001
**III trimester**	*p* < 0.001	*p* < 0.001	—	*p* = 0.469	*p* < 0.001
**Postpartum**	*p* < 0.001	*p* < 0.001	*p* = 0.469	—	*p* < 0.001
**Control**	*p* = 0.045	*p* < 0.001	*p* < 0.001	*p* < 0.001	—
**(I-III trim.) vs. Control**				*p* < 0.001	
**(I trim-post.) vs. Control**				*p* < 0.001	

SD—Standard Deviation, Control—Control Group.

## Data Availability

The data presented in this study are available on request from the corresponding author.

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
