# Peer review of "Tpeak-Tend Interval during Pregnancy and Postpartum"

_ijerph, 2022, doi:10.3390/ijerph191912638_

Round 1

Reviewer 1 Report

Few references are to be added

1. Tamirisa K, Elkayam U, Briller J et al. Arrhythmias in pregnancy.J Am Coll Cardiol EP. 2022 Jan, 8(1)  120-135.

2. Yilmaz M,Kayancicek H, Gozel N et al. Spotlights on some electrocardiographic paradigms: How should we evaluate normal reference valuesof Tp-Te interval,Tp Te dispersion and Tp-Te/QT ratio? Adv Clin Exp Med. 2020;29(9):1091-1099. doi:10.17219/acem/117684

3. Achmad C, Iqbal M, Karwiki G et al. T-Peak to T-End improvements after beta blocker administration in peripartum cardiomyopathy patients, Cardiol Res. 2020;11(3):185-191

Patients presenting in first trimester should have been followed up with ECG in each trimester, postpartum and after 6 weeks postpartum,  giving a proper study material and more authentic data. In stead authors have taken cross sectional data, Tp Te value may differ in different subjects. 

Reviewer 2 Report

I have read with great interest the manuscript entitled ‘Tpeak-Tend interval during pregnancy and postpartum’ submitted in ‘International Journal of Environmental Research and Public Health’ as an original article.

In this single-center prospective study, the Authors investigated the TpTe interval (Tpeak-Tend) in pregnant women that were in the first, second and third trimester as well as in post-partum period. The study group consisted of 128 women in pregnancy or postpartum period and a control group of 32 non-pregnant women. A standard 12-lead ECG (electrocardiograph) for evaluation of TpTe and QTc was performed in all patients. This study found an increase in the duration of the TpTe interval and QTc parameter during pregnancy and postpartum with the highest values in the postpartum period. TpTe interval increase was significantly higher compared to QTc increase during pregnancy and postpartum.

Overall, this is a very interesting original research article where Authors underscore the clinical significance and potential clinical implications of TpTe interval as a non-invasive electrocardiographic parameter.  This is an observational study and Authors provide an extensive review of the literature concerning the clinical significance of TpTe interval. The Authors should commend on the following comments:

1.      During normal pregnancies we may observe ECG changes such as reduction in the mean values of PR interval, sinus tachycardia, left axis deviation, inverted or flattened T waves and a Q wave in lead DIII. Heart rate is also found to increase progressively throughout the pregnancy, reaching a peak during the third trimester. Could changes in TpTe interval be attributed to normal changes occurring in pregnancy?

2.      The Authors conclude that ‘TpTe interval may identify women at risk of a potential ventricular arrhythmia enabling the early implementation of therapeutic and preventive measures’. However, (this is a great limitation of the study that the Authors need to acknowledge), changes of TpTe interval were not associated with any clinical outcome or measure of arrythmia burden. Thus, the Authors should conclude that further studies are warranted in order to see the clinical significance of this ECG finding.

3.      Another limitation of this study that the Authors should acknowledge is that ECG measurements were made manually.

4.      It would be interesting if the Authors could add echocardiography findings in the corresponding ECG changes.

Round 2

Reviewer 2 Report

The revised version of the manuscript includes satisfactory answers and comments to queries.